



# Quantification and assessment of methane emissions from offshore oil and gas facilities on the Norwegian Continental Shelf

Amy Foulds[1], Grant Allen[1], Jacob T. Shaw[1], Prudence Bateson[1], Patrick A. Barker[1], Langwen Huang[1], Joseph R. Pitt[1*], James D. Lee[2], Shona E. Wilde[2], Pamela Dominutti[2**], Ruth M. Purvis[2], David Lowry[3], James L. France[3], Rebecca E. Fisher[3], Alina Fiehn[4], Magdalena Pühl[4], Stéphane J. B. Bauguitte[5], Stephen A. Conley[6], Mackenzie L. Smith[6], Tom Lachlan-Cope[7], Ignacio Pisso[8], Stefan Schwietzke[9]

[1]Department of Earth and Environmental Sciences, University of Manchester, Oxford Road, Manchester, M13 9PL, United Kingdom
[*]Now at School of Chemistry, University of Bristol, Cantock's Close, Bristol, BS8 1TS, United Kingdom
[2]Wolfson Atmospheric Chemistry Laboratories, Department of Chemistry, University of York, Heslington, York, YO10 5DD, United Kingdom
[**]Now at Laboratoire de Météorologie Physique, Université Clermont Auvergne, Clermont-Ferrrand, 63000, France
[3]Department of Earth Sciences, Royal Holloway, University of London, Egham, Surrey, TW20 0EX, United Kingdom
[4]Deutsches Zentrum für Luft- und Raumfahrt (DLR), Institut für Atmosphäre, Oberpfaffenhofen, Germany
[5]FAAM Airborne Laboratory, National Centre for Atmospheric Sciences, Building 146, College Road, Cranfield, MK43 0AL, United Kingdom
[6]Scientific Aviation, Inc., 3335 Airport Road Suite B, Colorado 80301, United States
[7]British Antarctic Survey, Natural Environment Research Council, Cambridge, CB3 0ET, United Kingdom
[8]Norwegian Institute for Air Research (NILU), Kjeller, Norway
[9]Evironmental Defense Fund, Berlin, Germany

*Correspondence to*: Dr Amy Foulds (amy.foulds@manchester.ac.uk)

**Abstract.** The oil and gas (O&G) sector is a significant source of methane ($CH_4$) emissions. Quantifying these emissions remains challenging, with many studies highlighting discrepancies between measurements and inventory-based estimates. In this study, we present $CH_4$ emission fluxes from 21 offshore O&G facilities collected in 10 O&G fields over two regions of the Norwegian Continental Shelf in 2019. Emissions of $CH_4$ derived from measurements during 13 aircraft surveys were found to range from 2.6 to 1200 t year$^{-1}$ (with a mean of 211 t year$^{-1}$ across all 21 facilities). Comparing this with aggregated operator-reported facility emissions for 2019, we found excellent agreement (within $1\sigma$ uncertainty), with mean aircraft-measured fluxes 16% lower than those reported by operators. We also compared aircraft-derived fluxes with facility fluxes extracted from a global gridded fossil fuel $CH_4$ emission inventory compiled for 2016. We found that the measured emissions were 42% larger than the inventory for the area covered by this study, for the 21 facilities surveyed (in aggregate). We interpret this large discrepancy not to reflect a systematic error in the operator-reported emissions, which agree with measurements, but rather the representivity of the global inventory due to the methodology used to construct it and the fact that the inventory was compiled for 2016 (and thus not representative of emissions in 2019). This highlights the need for timely and up-to-date inventories for use in research and policy. The variable nature of $CH_4$ emissions from individual facilities requires knowledge of facility operational status during measurements for data to be useful in prioritizing targeted emission mitigation solutions. Surveys of individual facilities may always require this. However, for field-aggregated emissions, our results show that an accurate estimate of total field-level emissions simply requires a sufficiently large and representative sample of facilities, to yield meaningful comparisons and flux statistics, irrespective of operational status information. In summary, this study demonstrates the importance and accuracy of detailed, facility-level emission accounting and reporting by operators and the use of measurement approaches to validate bottom-up accounting.



# 1 Introduction

Concentrations of atmospheric methane ($CH_4$) have been increasing since 1850, with particularly rapid annual growth rates of over 5 ppb $yr^{-1}$ observed from 2014 to 2017 (Nisbet et al., 2019). With a radiative forcing of approximately 0.5 $Wm^{-2}$ (Prather et al., 2001) and a global warming potential 84 times that of $CO_2$ over a 20-year period (Myhre et al., 2013), $CH_4$ is the second-most important greenhouse gas. $CH_4$ emissions reduction and mitigation strategies could aid the attainment of climate targets set in the UNFCCC Paris Agreement (Nisbet et al., 2020). In order to inform and direct such efforts, an accurate understanding of the nature and magnitude of anthropogenic and natural sources of $CH_4$ is essential.

Emissions from the oil and gas (O&G) sector are estimated to account for approximately 22% of global anthropogenic $CH_4$ emissions (80 Tg $year^{-1}$), though this remains highly uncertain, with estimates ranging from 68 to 92 Tg $year^{-1}$ (Saunois et al., 2020). This can be partly attributed to the fact that O&G emissions are associated with a wide range of variable and episodic activities such as minor failures in engineering (Zavala-Araiza et al., 2017), flaring (combustion of the gas), controlled cold venting (discharge of unburned gases into the atmosphere) and other fugitive processes. Large but rare, unexpected leaks can also result in significant releases to the atmosphere (Ryerson et al., 2012, Conley et al., 2016; Lee et al., 2018).

There have been limited numbers of studies focussed on emissions from offshore O&G production, relative to onshore facilities (EIA, 2016a). The current quantification of emissions from offshore facilities therefore often relies on bottom-up approaches that use activity data and emission factors to derive emissions from a sub-set of sources, and extrapolation to estimate a total emission. However, emission factor calculations rely on representative knowledge of all emission sources, with the potential for systematic error. Top-down emission estimates, such as direct measurements of atmospheric mixing ratios downwind of a source or group of sources, can help to improve bottom-up inventory estimates, which in turn can more meaningfully inform emission mitigation and climate policy. However, the relatively small number of studies on offshore emissions means that there has been little independent data to validate reported emissions. The studies that have taken place have consistently reported inventory underestimates of $CH_4$ and non-methane volatile organic compounds (NMVOCs) from O&G extraction (Xiao et al., 2008; Pétron et al., 2012; Gorchov Negron et al., 2020).

Recently, ship-based campaigns have investigated $CH_4$ emissions from offshore facilities, including the Gulf of Mexico (Yacovitch et al., 2020) and the North Sea (Riddick et al., 2019). Yacovitch et al. (2020) reported $CH_4$ emission fluxes in the range of 0 to 190 kg $h^{-1}$ for 103 offshore facilities in the Gulf of Mexico region. Riddick et al. (2019) investigated $CH_4$ emissions from eight offshore facilities in UK part of the North Sea and reported leakage of $CH_4$ gas from all facilities sampled during normal operations, with a higher measured collective emission compared with estimates from the UK National Atmospheric Emissions Inventory (NAEI) (0.19% and 0.13%, respectively). Results from the Riddick et al., (2019) study emphasised a need for further research to accurately determine $CH_4$ leakages from offshore O&G facilities, and to include these in emission inventories. As part of the ACCESS campaign, Roiger et al. (2015) also highlighted the impact of offshore O&G facility emissions on local air quality, including nitrogen oxide ($NO_x$) emissions and tropospheric ozone ($O_3$) formation.

Gorchov Negron et al. (2020) derived facility-level $CH_4$ emissions from multiple offshore facilities in the Gulf of Mexico using aircraft observations. These were used alongside production data and inventory estimates to compile an aerial measurement based $CH_4$ emission inventory for the Gulf of Mexico. The inventory was separated into three source categories (producing facilities, non-producing facilities and minor sources, and largest shallow water facilities), with each category applying a different emission estimation approach. Comparisons with the USA Environmental Protection Agency greenhouse gas inventory showed that measured $CH_4$ emissions were consistent for deep water but were a factor of two higher for shallow water facilities. Gorchov Negron et al (2020) attributed this discrepancy to incomplete platform counts and discrepancies in the emission factors used in the inventory. In contrast, Zavala-Araiza et al. (2021) reported airborne measurements of $CH_4$ emissions from offshore facilities in the Sureste Basin, Mexico, which were found to be an order of magnitude lower than the Mexican greenhouse gas inventory.



As part of the United Nations Climate and Clean Air Coalition (CCAC) objective to quantify global $CH_4$ emissions from oil and gas facilities, this study quantifies $CH_4$ emissions from active O&G facilities on the Norwegian Continental Shelf using a Lagrangian mass balancing approach, as outlined in France et al. (2020). We report measurements of $CH_4$ mixing ratios and fluxes sampled by two research aircraft downwind of 21 emitting facilities (out of 25 facilities surveyed) during 13 flights in July and August 2019. The FLEXPART dispersion model was used to confirm the facility origin of sampled $CH_4$ plumes.

Comparisons are made with operator-supplied annualised emissions and daily activity data from individual facilities in order to identify agreements or discrepancies, as well as to evaluate the efficacy of emissions reporting procedures within the areas of the Norwegian Continental Shelf covered by this study. In particular, comparison with daily reported activity data is key when variable or episodic sources are present. Emission estimates from an annualised global inventory (Scarpelli et al., 2020) are also compared against measured data, to provide insight into the relative accuracy of a hierarchy of emissions accounting

approaches.

   In Sect. 2, we outline the details of the research aircraft, instrumentation and sampling strategies employed to survey emissions from O&G facilities on the Norwegian Continental Shelf. In Sect. 3, we describe the methods used to derive $CH_4$ fluxes from individual facilities and the uncertainties implicit to the mass balance method. In Sect. 4, we discuss the calculated facility-level flux results and compare them to estimates from both a global inventory and operator-reported emissions and activity

data. In Sect. 4, we also discuss the relevance of platform operational data and $CH_4$ loss rate calculations and provide an outlook for continued research in this field.

**2. Methods**

   In this section, we describe the flight surveys, aircraft platforms and instrumentation used to record measurements discussed in Sect. 4 and also describe the use of dispersion modelling for source attribution.

**2.1 FAAM BAe-146 research aircraft**

   Three flights (labelled C191, C193 and C197) were conducted by the UK's Facility for Airborne Atmospheric Measurement (FAAM) BAe-146 atmospheric research aircraft. Information regarding the full aircraft scientific payload can be found in Palmer et al. (2018). Here, we summarise the details of the measurements relevant to this study.

   Dry mole fractions of $CO_2$ and $CH_4$ were measured using a cavity-enhanced absorption spectrometer (Fast Greenhouse Gas

Analyzer (FGGA); Los Gatos Research, USA), sampling air through a window-mounted rear-facing chemistry inlet. A full description of the operation of the FGGA, along with its modification for measurements onboard the FAAM aircraft is reported by O'Shea et al. (2013). Raw data measured by the FGGA was corrected for small effects associated with water vapour dilution and spectroscopic error and calibrated using a three-point reference gas approach (high, low and target concentrations). Calibrations were performed approximately hourly in-flight using calibration gas cylinders traceable to the WMO-X2007 scale

(Tans et al. 2009) and WMO-X2004A scale (Dlugokencky et al., 2005) for $CO_2$ and $CH_4$, respectively. A target reference gas cylinder containing $CH_4$ with a mole fraction approximately half-way between that of the hourly high-and-low calibrations (equal to 1879.58 ppb) was also sampled hourly to quantify small sources of instrumental temporal drift and non-linearity and thereby to define measurement error. For a full description of the water vapour correction, calibration regime and measurement validation, see O'Shea et al. (2013). The representative one standard deviation calibration measurement uncertainties were

3.62 ppb for $CH_4$ and 0.84 ppm for $CO_2$ at a sample rate of 10 Hz. The limit of detection of high precision optical cavity instruments such as those used on all platforms in this study is well below the atmospheric background concentrations of $CH_4$ and $CO_2$. Therefore, flux calculations are not limited by the precision of such instruments, but rather, by the environmental conditions at the time of the survey (see Sect. 3.1 and France et al. (2020) for a full discussion). Using the methods, platforms and instruments described in this paper, we estimate that a flux at the order of 2 kg h$^{-1}$ represents a typical flux limit of detection



for the range of conditions experienced in the fieldwork presented in this paper. However, as discussed, the true limit of detection will depend on the environmental conditions at the time of each survey.

Thermodynamic measurements were used to diagnose boundary layer mixing processes (Sect. 3). Ambient temperature was measured using a Rosemount 102AL sensor, which has an overall measurement uncertainty of ±0.3 K and 95% confidence. Measurements of static air pressure were recorded from pitot tubes along the aircraft, with an accuracy of ±0.5 hPa.

Measurements of 3-dimensional wind were made using a nose-mounted five-hole probe system described by Brown et al. (1983), with a horizontal wind measurement uncertainty of $< \pm 0.5$ ms$^{-1}$. A full description of the meteorological and thermodynamic instrumentation on board the FAAM aircraft can be found in Petersen and Renfrew (2009).

### 2.2 Scientific Aviation Mooney aircraft

The Scientific Aviation airborne measurement platform consists of a single engine propeller Mooney aircraft, outfitted with

trace gas instrumentation. Air was continuously drawn through rearward-facing inlets installed on the aircraft wing and delivered to instruments in the aircraft cabin through stainless steel or Teflon tubing. $CH_4$, $CO_2$, and water vapour ($H_2O$) were measured by wavelength-scanned cavity ring-down spectroscopy in a Picarro model G2301-f detector. Precision of the G2301-f $CH_4$ measurement was < 1 ppb at 0.5 Hz. Ambient temperature and relative humidity were measured by a wing mounted Vaisala HMP60 probe. Aircraft position was measured using a Hemisphere high-precision differential GPS system and wind

speed and direction were calculated according to Conley et al. (2014).

### 2.3 Flight sampling and study area

Over the course of this campaign, 21 offshore O&G facilities were surveyed by both aircraft plus repeats at some facilities (see details below, 34 surveys in total).

### 2.3.1 FAAM flights

The FAAM research aircraft conducted three regional flight surveys of two regions on the Norwegian Continental Shelf in July and August 2019, as part of the "Methane Observations and Yearly Assessments" (MOYA) project, funded jointly by the Natural Environment Research Council (NERC) and the United Nations Environment Programme: Climate and Clean Air Coalition (UNEP CCAC). Figure 1 illustrates the two regions surveyed by the flights, along with O&G facilities in the area.

During each of the FAAM survey flights, emissions from between two and four facilities were detected. These facilities

were identified as the sources of the observed $CH_4$ plumes, using on-board wind direction and $CH_4$ measurements, alongside the GPS coordinates of the facilities. The atmospheric dispersion model, FLEXPART, was also used to aid source identification (Sect. 2.4).





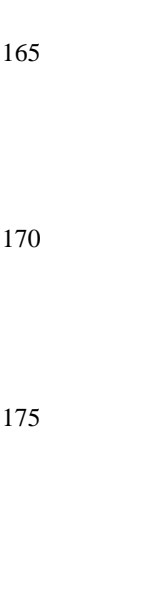

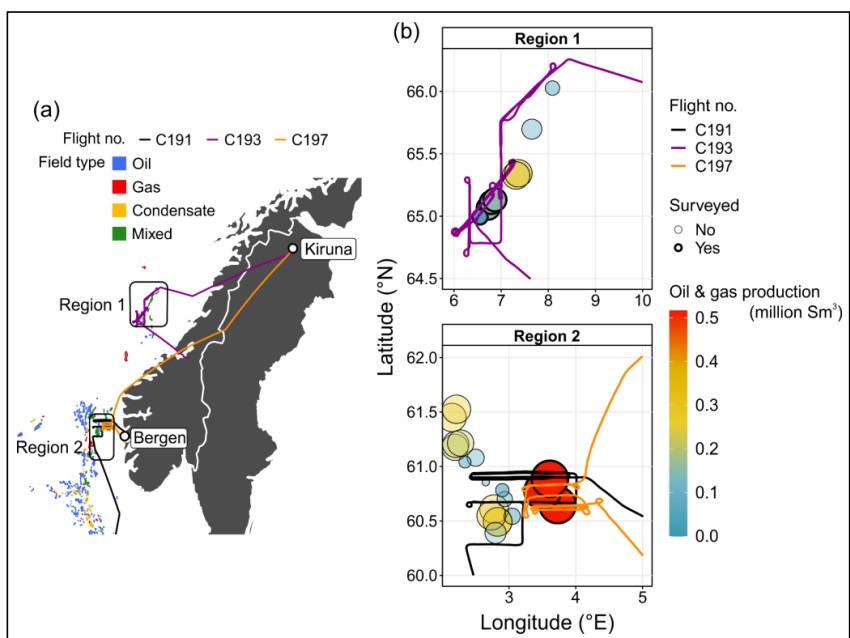

Figure 1. (a) Location of offshore fields on the Norwegian Continental Shelf and FAAM aircraft survey patterns (as coloured tracks). Each symbol represents an offshore field, coloured by extraction product (b) Map of the FAAM flight tracks and locations of active O&G facilities in the two regions. Each circle represents a distinct facility, sized and coloured according to the reported O&G production in 2019 (Norwegian Pet

roleum Directorate, 2021), with the bold, opaque circles denoting the facilities surveyed in this study. For ease of illustration, the size of each point is also scaled for the relative oil and gas production. However, the colour legend reflects the annual oil and gas production.

The two regions were selected due to the large amount of oil and gas produced by facilities in each region, as seen in Figure 1. Flight C191, in region 2, sampled between 134 and 370 m above sea level (masl) with straight-and-level transects at 150 masl upwind of the facilities to provide a representative background measurement. Repeated reciprocal runs at varying altitudes within the boundary layer were carried out downwind of sources to detect and characterise emission plumes. Flights C193 and C197 were conducted in regions 1 and 2, respectively. These flights involved two sets of vertically stacked transects at various altitudes. In flight C193, these transects ranged from 124 to 606 masl with altitudes in flight C197 ranging from 103 to 308 masl. All three FAAM flights were conducted when the cloud base exceeded 300 masl, to ensure good visibility and allow for low altitude sampling. Across the three flights, the number of stacked transects ranged from 7 to 14, at between 50 and 100 m spacing. See Appendix Fig. B2 for an example altitude-longitude projection of the stacked legs flown in flight C193. All three FAAM flights were conducted when the cloud base exceeded 300 masl, to ensure good visibility and allow for low altitude sampling. There was no contact with the operators prior to or during the flights, where the operators were informed about the measurements.

### 2.3.2 Scientific Aviation flights

Concentric closed flight laps were flown around each target site (individual facility), beginning at the lowest safe flight altitude (20 to 190 masl) to an altitude exceeding the observed maximum emission plume height (typically 100 to 800 masl), creating a virtual sampling cylinder incorporating both upwind background and downwind plume measurement. The number of laps varied for each facility surveyed, typically ranging between 5 and 25. See Appendix Fig. B3 for an example plot of one of





these surveys. The highest altitude flown for each site was determined by the absence of significant upwind/downwind variability in the trace gas signal measured onboard the aircraft (i.e., no downwind $CH_4$ enhancements were observed). The downwind lateral distance at which the plume was intercepted by the aircraft was typically 1-2 km.

The measurement sites were selected based on proximity to Bergen Airport, Norway, with facilities within approximately 200 km being investigated. Operators of target sites were informed of measurements on the common frequency for the local area during the flight itself. All airborne measurements were conducted under visual flight rules (VFR) flight conditions, meaning the aircraft was not flying in clouds, fog, or low-visibility areas. This was done to ensure that a safe flying distance was maintained between the measured facilities and the sea surface.

Between two and eight facilities were surveyed on each of ten survey flights conducted in August and September 2019. Over the course of the campaign, 21 O&G facilities were investigated (17 offshore facilities reported in Figure 2), with repeated surveys of eight facilities over several days. The locations of the offshore O&G facilities surveyed during the Scientific Aviation flights are shown in Figure 2.


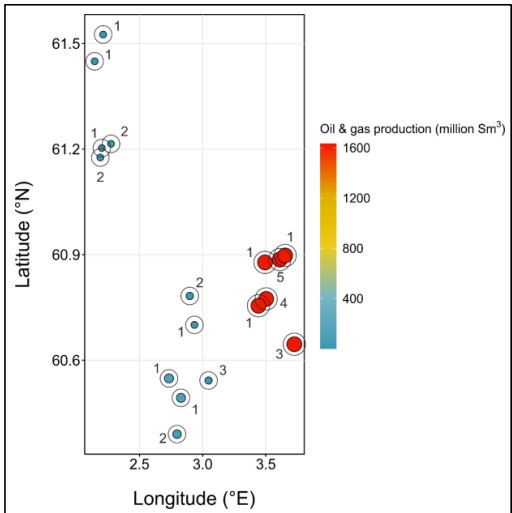



Figure 2. Location of the offshore O&G facilities sampled by the Scientific Aviation aircraft. Each platform is coloured and sized according to respective O&G production for 2019 (Norwegian Petroleum Directorate, 2021). Circles around each platform are used to illustrate the concentric flight laps conducted by the aircraft during sampling, with the numbers denoting the number of times each facility was surveyed.


### 2.4 Atmospheric dispersion model configuration

FLEXPART ("FLEXible PARTicle dispersion model) is a Lagrangian dispersion model. FLEXPART was used to model the

$CH_4$ emission plumes for target facilities. Backward plumes (or footprints) were also simulated, based on measured $CH_4$ data, to confirm the source facility of origin based on the locations of plumes sampled downwind during FAAM surveys. Source attribution was not necessary for Scientific Aviation surveys by virtue of the close proximity cylindrical sampling permitted by the smaller Mooney aircraft. FLEXPART simulates the Lagrangian trajectories of a large number of particles in the atmosphere. These particles, tracked forward or backwards in time, were driven by Eulerian wind fields produced by the

European Centre for Medium-Range Weather Forecast (ECMWF) with 0.1° horizontal resolution, and 137 vertical levels from the surface to approximately 80 km. The domain used was [0.0°E 12.0°E; 57.0°N 68.0°N].



Two different sets of simulations were performed: backwards for source identification associated with individual airborne measurements, and forward to aid constraint of the maximum plume mixing height used for flux quantification (as described in Sect. 3). Backward plumes (or footprints) for every discrete measurement point were calculated along the flight tracks of

FAAM flights C191, C193 and C197. For the backward simulations, the output grid resolution was 0.01° x 0.01° (~ 1 km at the equator) in the horizontal and 10 m in the vertical. Trajectories of 20,000 particles were calculated per individual measurement point. The footprint determined by the model was used to provide an estimated contribution from the facilities in question to the measured $CH_4$ enhancement at the point of measurement. This was then used to attribute the individual $CH_4$ plumes to specific facilities, based on the co-location of measured plumes. In addition, forward FLEXPART simulations were

run, with output produced at the same resolution as the backward simulations, in order to estimate the maximum mixing height of the forward plumes emitted from the facilities on the Norwegian Continental Shelf. Example model data plots from both of these types of dispersion simulation can be found in the Appendix A. A detailed description of the FLEXPART model and its components can be found in Pisso et al. (2019).

### 2.5 Reported emission data sources

**2.5.1 Annualised $CH_4$ emission & activity data from platform operators**

In Norway, facility-level reporting of offshore O&G $CH_4$ emissions is based on calculations at the source-level using recommended guidelines (Norwegian Oil and Gas Association, 2018a), the results of which are then published (Norwegian Oil and Gas Association, 2021). In this study, an inventory of existing O&G related facilities in the study area as well as activity data including O&G production statistics and facility functions were obtained from public data sources

(www.norskoljeoggass.no and www.norskeutslipp.no). Additional data related to temporary facility activities such as flaring status or compressor ramp up for the days of the aerial surveys were provided via direct communication with the respective operators. Operators of facilities on the Norwegian Continental Shelf are required to submit annual $CH_4$ emissions data to the Norwegian Environment Agency annually. The $CH_4$ emissions for individual sources and sub-sources (e.g. primary vent seals for centrifugal compressors and incomplete combustion in flares). The basis for the reporting is a

project led by the Norwegian Environment Agency between 2014 and 2016, which focuses on direct $CH_4$ emissions from O&G production activities on the Norwegian Continental Shelf (Husdal et al., 2019). All installations were subject to a detailed mapping of all potential sources of direct $CH_4$ emissions, and updated methodologies for quantifying emissions at the source/sub-source level were established based on best available techniques. The industry was an active participant in the project, and detailed recommended guidelines for emission and discharge reporting were established (Norwegian Oil and Gas

Association, 2019). This was followed by a handbook for quantifying direct $CH_4$ and NMVOC emissions (Norwegian Oil and Gas Association, 2018b) and a guideline for the quantification of small leaks and fugitive emissions (Norwegian Oil and Gas Association, 2021b). The $CH_4$ reporting methodology on the Norwegian Continental Shelf is amongst the most advanced in the O&G industry globally, as each individual $CH_4$ emission source/sub-source is configured at each installation (i.e. if gas is recycled, flared or cold vented). The detailed reporting associated with each facility is publicly available (Norwegian Oil and

Gas Association, 2021c).

### 2.5.2 Global inventory of $CH_4$ emissions from oil and gas exploitation

Measured $CH_4$ emissions from individual facilities were compared with a regional sample of a global, gridded inventory of $CH_4$ emissions from oil, gas and coal exploitation with a resolution of 0.1° by 0.1° for the year 2016 (Scarpelli et al., 2020). The gridded inventory resolves contributions from individual subsectors (exploration, production, transport, refining) and from

specific processes (flaring, venting, leakage). National emissions for each of these subsectors and processes were routinely compiled from UNFCCC national reported emissions using IPCC Tier 1 methods (IPCC, 2006). Such methods apply default



emission factors (not country-specific) and activity data which are limited to national O&G activity statistics. These national emissions are then spatially allocated on the inventory's 0.1° by 0.1° grid across specific O&G infrastructure, in order to derive spatially aggregated emission estimates for infrastructure in each grid cell. The inventory therefore acts as a spatially

downscaled representation of these UNFCCC reports. Higher tier IPCC approaches are assumed to be much more rigorous and detailed. For example, Tier 2 approaches use country-specific emission factors. Tier 3 approaches apply a rigorous bottom-up assessment of emissions by primary source type (venting, flaring) using data reported by individual facilities (IPCC, 2006). This is a much more detailed and extensive process for compiling emissions. However, not all nations or facilities collect or report such data, meaning that it would not be an effective or consistent way to derive emissions for a global inventory. As

discussed in Sect. 2.5.1, facility-level reporting of offshore O&G $CH_4$ emissions in Norway is based on calculations at the source-level using recommended guidelines (Norwegian Oil and Gas Association, 2018). In this context, the comparisons made in this study represent a comparison with a spatially downscaled estimation approach (Scarpelli et al. (2020) inventory) and the more detailed quantification approach used by O&G facility operators (facility-level reports).

Annualised gridded emission fields for O&G platforms for the year 2016 were downloaded from the Harvard Dataverse

(Scarpelli et al., 2019). Equivalent inventory data for 2019 was not available at the time of the study. This is often a problem for inventory comparisons, as some inventories are not updated in real time, which can impact the accuracy of comparisons if changes in infrastructure may be expected in the intervening time. We include the comparison here as an illustration of this challenge. $CH_4$ emissions associated with the platforms of interest were extracted, using their geographical coordinates to identify the corresponding grid cell and $CH_4$ emission in the inventory.

**3 Flux analysis methodology**

In this section, we describe the flux quantification method applied to sampling from the FAAM and Mooney aircraft surveys and describe the quantification of flux uncertainty.

**3.1 Aircraft mass balance**

Fluxes can be quantified using mass balance approaches. For such approaches to be feasible, observations are typically made

upwind of the source region, to establish concentrations in a background location. Downwind observations are then conducted, allowing the determination of the net enhancement attributed to the source region. Lagrangian mass balance flux quantification typically requires meteorological conditions where the wind field can be assumed (and measured) to be relatively invariant over the spatial scales of plume sampling for a target emitter (Cambaliza et al., 2014; Pitt et al., 2019; Fiehn et al., 2020). Often, it is assumed that the plume is vertically well-mixed within some layer (usually the planetary boundary layer). The

vertical mixing assumption also requires that measurements are taken sufficiently downwind of the emission source so as to have had time to fully mix. The aircraft mass balance approach used in this study has been used to derive fluxes of trace gases from large area sources, such as agriculture, oil and gas fields and cities (e.g. White et al., 1976; Wratt et al., 2001; O'Shea et al., 2014; Peischl et al., 2016; Pitt et al., 2019), but has also been used for individual O&G facilities (e.g. Lee et al., 2018; Guha et al., 2020).

**3.1.1 FAAM flights**

The emission fluxes presented in Sect. 4 were calculated from the FAAM survey flight data using Eq. (1):

$$F = \int_0^{z_{max}} \int_A^B (C_{ij} - C_0)\, n_{air} U_{\perp ij}\, dx\, dz \tag{1}$$

where, $F$ (g s⁻¹) is the flux for the emission source, $C_{ij}$ is the dry mole fraction of $CH_4$ at each point in the plume, $C_0$ is the

representative background dry mole fraction of $CH_4$, $n_{air}$ is the molar air density and $U_{\perp ij}$ is the wind speed perpendicular to





the reference measurement sampling plane. For the flux calculations in this study, the atmosphere was divided into discrete vertical layers, based on the mean altitudes of aircraft transects for each facility survey. The mean concentrations within each observed $CH_4$ plume were used to calculate flux individually for each layer and summed across all layers to obtain total flux.

Representative background $CH_4$ mole fractions were determined for each layer using the 50 neighbouring 10 Hz measurements either side of the observed plume. The average $CH_4$ enhancement above this background was calculated for each observed plume. The perpendicular wind speed was calculated as the average wind vector component perpendicular to each flight transect. Plume mixing altitude was calculated as the distance between the sea surface and either the point at which a plume was no longer observed in measured data, or the height of the mixed layer as diagnosed from FLEXPART forward modelling

or the nearest available potential temperature profile measured by the aircraft. In the absence of a direct measurement of plume mixing height, where the boundary layer height or FLEXPART model mixing was used to define the plume mixing height, the difference between the nearest altitude where a plume was measured, and the assumed mixing height, was used to define a quantifiable vertical mixing uncertainty used in flux error propagation (see Sect. 3.2). In summary, for surveys where the plume top could not be directly constrained by measurement, any assumed vertical mixing was conservatively accounted for

within the quoted flux uncertainty reported in Sect. 4.

### 3.1.2 Scientific Aviation flights

A variant of the Lagrangian mass balance method, utilising Gauss' Theorem and suited to the orbital sampling conducted by the Mooney aircraft, was used to derive $CH_4$ fluxes from the Scientific Aviation flight surveys. Gauss' theorem was used to estimate $CH_4$ flux through the virtual cylinder created by flying concentric circles around an individual platform. This theorem

equates the volume integral of the source (e.g. platform) to a surface integral of the trace mass flux which is normal to the surface of a cylinder. The volume integral was converted to a surface integral, which was used to calculate the horizontal mass flow of $CH_4$ across the cylinder's surface plane. All other flux parameters in Eq. 1 were calculated in the same way as for the FAAM flight surveys. A full description of this emission quantification method can be found in Conley et al. (2017).

### 3.2 Flux uncertainties

### 3.2.1 FAAM flights

The uncertainty in the measured flux was determined using a similar method to that used by O'Shea et al. (2014). This involves propagating the measured uncertainties associated with the individual terms in Eq. (1), including the uncertainty in the observed $CH_4$ enhancement, the natural (measured) variability of the wind field, and any uncertainty in the plume mixing height. Instrumental uncertainties associated with the FGGA were calculated to be negligible in comparison to those associated

with the wind field and plume mixing height but are implicitly accounted for within the measured variability (and hence uncertainty) in the background concentration.

        Non-correlated, random uncertainties (wind and background variability) were summed in quadrature and calculated as an uncertainty for each altitude layer. These were then summed for all altitude layers to derive an overall random uncertainty in the corresponding total flux. The systematic uncertainty in the plume mixing height (described in Sect. 3.1.1) was then added

to the random error to obtain the total uncertainty in the flux reported for each facility.

### 3.2.2 Scientific Aviation flights

The uncertainties in emission flux (reported as a one standard deviation uncertainty) were calculated as follows, and analogous to those calculated for FAAM survey data. Firstly, the statistical (random) uncertainty in the wind field and the $CH_4$





measurement from the Picarro instrument were summed in quadrature, in order to obtain uncertainty in the horizontal flux for
each concentric lap. The horizontal fluxes were then binned in altitude layers, and the uncertainties of the horizontal fluxes in
that bin were summed in quadrature along with the standard deviation of the flux estimates for each layer. The uncertainties
in each bin were added in quadrature to obtain the final error estimate for the total flux measurement for each individual survey.

Where multiple surveys were conducted over several days, this was taken into consideration when calculating the overall
uncertainty for each facility. The relative error for each survey was calculated. These were then averaged to give a mean
uncertainty over all surveys for each facility. The mean relative uncertainty was then multiplied by the average $CH_4$ flux, to
obtain a mean-weighted uncertainty in the $CH_4$ flux for each facility.

**4 Results & discussion**

In this section, we report the measured fluxes for each facility and compare with inventory and facility-level activity data.
Details about the observational data from the FAAM and Scientific Aviation flight surveys, and the application of the mass
balance approach can be found in Appendix B.

**4.1 Measured flux uncertainties**

Uncertainties in flux are a function of sampling density, background variability, wind conditions, as well as the instrumental
uncertainty (France et al., 2020). Combined uncertainties associated with background and wind variability were observed to
be less than 10% in the FAAM flight surveys of this case study. The largest source of flux uncertainty in the FAAM flight
surveys was found to be in the plume mixing height (typically accounting for more than 90%). As discussed in Sect. 3.1.1.,
this was calculated as either the height at which a plume ($CH_4$ enhancement) was no longer observed downwind, or in the
absence of a vertical measurement constraint, as the nearest available measured thermodynamic boundary layer height as a
proxy for maximum possible mixing. The vertical plume was more constrained by the Scientific Aviation flight patterns due
to the dense vertical sampling made possible by the more agile, smaller Mooney aircraft, reflected by the smaller flux
uncertainties in the Scientific Aviation surveys (see Table 1). There is also some uncertainty if the bottom of the plume cannot
be sampled. This is captured in the uncertainties reported for all flights and represents an inherent limitation of all aircraft
surveys. By way of forward guidance, an optimal sampling design (to minimize flux uncertainty) therefore involves repeated
sampling at many altitudes around a target of interest, ensuring that the top of any plume is directly measured.

**4.2 Flux comparisons with a global inventory and facility-level reported data**

This study involves direct comparisons of the measured $CH_4$ fluxes with those reported by facility operators and global
emission inventory estimates. This requires temporal unit conversions of the measured data from g s⁻¹ and kg h⁻¹ to t year⁻¹.
Scaling in this way is clearly not a robust comparison as it cannot account for any variability in day-to-day facility operations
throughout the year. Such day-to-day variability has also been observed and discussed in Tullos et al. (2021), whereby short-
duration $CH_4$ measurements were made at 33 dry-gas production sites in East Texas over the course of three weeks. This study
demonstrated that observations made at the same sites, within days of each other, could result in very different emission
estimates. However, as it is impractical to quantify the emissions from the facilities every day of the year, flight surveys
provide us with "snapshots" of the emissions, scaled to annualised data for direct comparisons, and yields insights into the
sources of any observed discrepancies, especially when comparing a large number of surveys and facilities in aggregate. This
annualised approach has been used to compare inventories with discrete measurement surveys of offshore O&G facilities, as
discussed in Sect. 1.





Figure 3 shows the spatially gridded CH$_4$ estimated emission data from the Scarpelli et al. (2020) global inventory for the Norwegian Continental Shelf. The estimated emissions shown represent those sourced from fuel exploitation (i.e., oil, gas and coal) for the year 2016. The highest estimated emissions in the area of interest range from approximately 1.6 to 2.0 t CH$_4$ year$^{-1}$

km$^{-2}$ and it is this data which is used to compare against the measured CH$_4$ fluxes from the aircraft surveys in this study.  We recognise that the emissions derived from this inventory are estimates for the individual facilities surveyed in this study, and do not reflect what is reported by operators. Inventory estimates such as this are not used as the basis for national emissions reporting.





Figure 3. Spatially gridded CH$_4$ emissions from fuel exploitation for the northern North Sea and Norwegian Sea (Scarpelli et al., 2020). Regions surveyed in this study are represented by the boxes labelled "Region 1" and "Region 2".

Figure 4a displays the measured CH$_4$ emission fluxes and corresponding spatially downscaled inventory estimates for all of the offshore O&G facilities surveyed in this study. The inventory contains significantly underestimated emissions for facility 2 (seen as the outlier in Figure 4a with an inventory flux ~ 60 t yr$^{-1}$), with measured CH$_4$ emission fluxes over a factor 20

higher, whilst also noting that the measured flux uncertainty was high. However, considering the low R$^2$ value (0.02) in Figure 4a, we emphasise that the intercepts and gradients calculated in this regression analysis are not meaningful, due to the high variability of agreements amongst the individual facilities. We include the result here to make this valuable point, which is to say that comparisons of surveys with spatially downscaled inventories may have limited value and require careful thought before drawing conclusions.

Figure 4b compares the measured emission flux with facility-level reported emissions, which have similarities with Tier 3 emission reporting. This figure shows a much closer agreement than that observed in Figure 4a, with an improvement in the number of facilities falling within the 95% confidence interval of the fitted regression between measured and reported emission flux. Figure 4b shows two outliers with reported fluxes of 270 and 780 t year$^{-1}$. These correspond to facilities 20 and 17, respectively, with significantly smaller measured emission fluxes. A near-zero measured emission flux was reported for facility

20. This is consistent with temporary inactivity, resulting from turbine maintenance on this O&G facility reported on the day of the flight survey. Correspondence with operators of facility 17 highlighted the fact that the cold vent is located in close proximity to the ignited flares, meaning that some CH$_4$ gas may be combusted as it passes near to the flare. This would not be implicitly accounted for in the reported fluxes for this facility, which assume that all cold-vented gas is emitted directly, without any combustion taking place.  Consequently, this could result in an under-bias in the reported emission fluxes, and





445 hence the observed discrepancy when compared with the measured emission fluxes for facility 17. The nature of cold venting and the potential for combustion therefore represents a potential problem for accurate $CH_4$ emission reporting.

  The regression in Figure 4b does not include reported emissions of zero, as shown in Figure 4a, as the two regression lines were found to be essentially identical. These results show that on aggregate, with a sufficient number of surveys, measurements are able to replicate the facility-level reported emissions, whilst also confirming that facility-level reporting procedures can

450 provide accurate emission estimates for incorporation into inventories. Facility 2 (the outlier seen in Figure 4a, discussed above) shows good agreement between operator-reported emissions and measured data, suggesting that facility-level reported flux for facility 2 is much more accurate than that represented by the inventory, and therefore that the observed difference between the measured and inventory emission estimates can be attributed to the emission calculation methodology applied in the inventory. This is consistent with the conclusions of other studies that have compared top-down measurements and global

455 inventories compiled using Tier 1 approach (Sect. 2.5.2; Gorchov Negron et al., 2020; Zavala-Araiza et al., 2021).

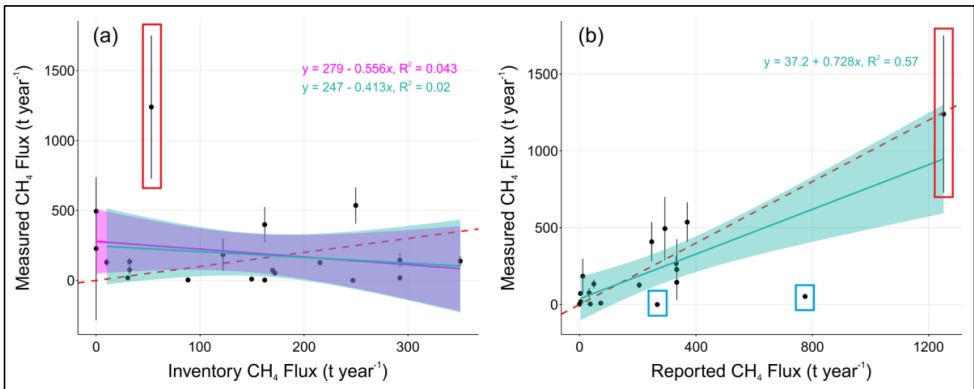

Figure 4. Comparison of measured $CH_4$ emission fluxes and a) corresponding estimates from the Scarpelli et al. (2020) global inventory, and b) corresponding operator reported emissions for each facility. The data point in the red box represents a particularly high measured flux of 1239.65 t year$^{-1}$. The data points in the blue boxes in Figure 4b) represent the two outliers, with near-zero measured fluxes. The green line represents a fitted linear regression through all data points and the magenta line represents a fitted regression which excludes inventory or reported emissions with a value of 0. This magenta line is not shown in Figure 4b, as the two regression lines were essentially identical. The font colours of the equations shown correspond to the colour of the respective regression line. The shaded regions of these lines correspond to the 95% confidence levels for the slopes. The red dashed line shows the 1:1 correspondence line for illustration.


  Table 1 compares the measured $CH_4$ emission fluxes with both annualised facility-level fluxes reported by respective facility operators (using similar approaches to those applied in IPCC Tier 3 emission quantification approaches), and corresponding emission estimates from the Scarpelli et al. (2020) global emission inventory (compiled using the IPCC Tier 1 approach). Due to commercial sensitivity, the platforms are arbitrarily labelled with a number in Table 1 and the operators are not identified.

480 Facilities 6 and 7 were surveyed separately by the aircraft. However, the reported emissions were grouped for the two facilities, which impedes our ability to directly compare to the reported flux for each facility separately. However, our observations show that facility 6 dominated emissions (400 t year$^{-1}$), relative to facility 7 (9.6 t year$^{-1}$). For individual facilities, there are notable large differences between the inventory estimates and the extrapolated measured emission fluxes, ranging from -41% (-88 t year$^{-1}$) to 2200% (1200 t year$^{-1}$) for facilities 5 and 2, respectively. This is expected to be associated with both the compilation

485 methodology of the inventory, whereby national emissions are downscaled to corresponding infrastructure, and the fact that the Scarpelli et al. (2020) inventory was compiled for the year 2016. The latter was due to equivalent inventory data for 2019





not being available at the time of this study, thus illustrating the challenge of inventory comparisons, with respect to infrastructure changes which may take place in the intervening time period between the inventory compilation and the surveys. Global inventories such as this do not have the granularity or detail compared with that provided by operator-reported data for

individual facilities (see Sect. 4.3). Such large differences between top-down methods and emission inventories have been reported previously (see Sect. 1). Gorchov Negron et al. (2020) compared regional airborne estimates of $CH_4$ emissions from offshore O&G facilities in the Gulf of Mexico with the USA Environmental Protection Agency greenhouse gas inventory, with measured $CH_4$ emissions found to be consistent for deep water but a factor of two higher for shallow water facilities.

Considering all facilities collectively, the measured fluxes were found to be 42% greater than the Scarpelli et al. (2020) emission inventory using Tier 1 methods. However, there is a much-improved agreement when comparing with the facility-level reported flux where measured fluxes are 16% lower than those reported. This aggregated comparison with facility-level reported data suggests that measurements and reported data agree within uncertainty, given a large enough sample size, and therefore we recommend that facility-level reporting is adopted more widely and used to compile more robust inventories of

$CH_4$ emissions. As discussed earlier, the Scarpelli et al. (2020) inventory was compiled for 2016 as equivalent 2019 data were unavailable at the time of this study. Using a scale factor, derived as a ratio between 2016 and 2019 total reported emissions data from the offshore fields (Norwegian Oil and Gas Association, 2021), we can proportionally scale 2016 inventory estimates to better represent 2019 when comparing measured emissions to the Scarpelli inventory. Repeating the analysis above, using the scaled inventory, we find that total measured emissions were 52% higher than the inventory for 2019. This further

highlights the limitation of comparisons with global inventories and their Tier 1 approach, and shows that a better agreement can be observed when comparing with a more specific inventory (e.g. facility-level reported emissions). Therefore, the poorer agreement between the measured fluxes and the Scarpelli inventory can be interpreted to reflect the representivity of the inventory, due to its construction methodology and the fact that it was compiled for 2016 (and thus, is not representative of emissions in 2019), rather than a systematic error in the operator-reported emissions, which agree with the measured fluxes.










Table 1. Summary of the measured $CH_4$ fluxes and comparison with respective emission data from the Scarpelli et al. (2020) global inventory and reported $CH_4$ emissions from the O&G facility operators. All measured, reported and Scarpelli inventory fluxes are quoted to two significant figures here. Flux uncertainties represent one standard deviation confidence intervals for each facility (see Sect. 3.2 for details).

| Facility I.D. | Research Aircraft | Number of Surveys | Measured $CH_4$ Flux and Uncertainty / t year$^{-1}$ | Reported $CH_4$ Flux (2019) / t year$^{-1}$ | Scarpelli Inventory $CH_4$ Flux (2016) / t year$^{-1}$ |
|---|---|---|---|---|---|
| 1 | FAAM | 1 | 490 ± 210 | 290 | 0 |
| 2[a] | FAAM | 1 | 1200 ± 510 | 1300 | 53 |
| 3 | FAAM | 1 | 230 ± 200 | 330 | 0 |
| 4 | FAAM & SA | 5 | 180 ± 110 | 11 | 120 |
| 5 | SA | 4 | 130 ± 20 | 210 | 220 |
| 6 | FAAM & SA | 7 | 400 ± 130 | 250[d] | 160 |
| 7 | SA | 1 | 9.6 ± 13 | 250[d] | 150 |
| 8 | SA | 1 | 72 ± 15 | 3.3[e] | 170 |
| 9 | SA | 1 | 2.6 ± 6.1 | 0[f] | 160 |
| 10 | SA | 3 | 540 ± 130 | 370 | 250 |
| 11 | SA | 1 | 18 ± 23 | 4.4 | 290 |
| 12 | SA | 2 | 150 ± 51 | 330 | 290 |
| 13 | SA | 1 | 140 ± 37 | 330[d] | 350 |
| 14 | SA | 1 | 130 ± 27 | 330[d] | 10 |
| 15 | SA | 1 | 76 ± 32 | 33 | 32 |
| 16 | SA | 1 | 18 ± 15 | 73 | 30 |
| 17 | SA | 1 | 53 ± 17 | 780 | 172 |
| 18 | SA | 1 | 3.5 ± 8.8 | 37 | 88 |
| 19 | SA | 2 | 130 ± 29 | 49 | 31 |
| 20 | SA | 2[c] | -0.88 ± 1.8[b] | 270 | 247 |

[a] Collective I.D. for two facilities, to coincide with grouping in inventory and reported estimates.
[b] The relatively low absolute mean flux with a negative sign is an artifact of minor upwind $CH_4$ contamination overwhelming
the downwind $CH_4$ enhancement. It is acknowledged that physical $CH_4$ emissions from this facility cannot be negative.
[c] Facility was surveyed twice. Only one measured flux is reported, as upwind contamination invalidated the second measurement.
[d] Both facilities were measured separately, but operator reports a combined estimate.
[e] Facility is a subsea manifold station. A drilling vessel was drilling at the same location at the time of surveying. The operator
reported that drilling was the main $CH_4$ source of >99% of $CH_4$ emissions and that $CH_4$ emissions will only occur during drilling.
[f] Operator does not report emissions as this facility was reported inactive during 2019.

**4.3 The relevance of platform operational data and $CH_4$ loss rate calculations**

Figure 5 displays a summary of facility-level $CH_4$ emission estimates including repeat measurements. Hourly emission rates were annually extrapolated for comparison with reported values. Panel (a) groups the facility IDs into three clusters. The first cluster (IDs 4-7) contains facilities for which measurements were available under both normal facilities operations and "other" operations. Other operations are defined as temporal deviations from the normal operations (based on operator reports received



upon request post-campaign), which may increase or decrease snapshot emission estimates relative to annualized inventories.

The second cluster (IDs 1-3, 10, 11, 13-17, 19) contains facilities for which measurements were available only under normal facility operations. The third cluster (IDs 8, 12, 18, 20) contains facilities for which measurements were available only under "other" facility operations. Panel (b) shows the total facility emissions, which is based on average facility emissions for repeated surveys. The first column represents only the first cluster from panel (a). The second column represents only the second cluster from panel (a). The third column represents all facilities from panel (a), i.e., a mix of normal and other

operations. The facility-level uncertainties shown by the error bars in Figure 5 were propagated in panel (b) using a Monte Carlo simulation, assuming normally distributed errors and independent samples. The latter is based on the fact that repeat sampling occurred on different days and individual platforms operate independently of one another.

As shown in Figure 5a, operator-reported facility-level, annualised emission rates agree with single survey measurements within uncertainties for 24% of the offshore surveys. However, for 76% of the surveys, reported emissions underestimate or

overestimate measured values at individual facilities independently of whether the facilities were surveyed under normal (continuous) operations or "other" operations. The list of other operations include: facility turnaround, turbine/compressor irregularities (such as lower than usual turbine load, compressor out of operation, or compressor ramp up and shutdown), reduced gas production or routing to a connected facility, increased flaring, and well drilling. The operation report descriptions thus suggest that "other" operations are expected lead to either increased or decreased emissions relative to the annual average

emissions. Indeed, the measurements confirm this expectation (reported emissions tend to underestimate measurements at facilities 4, 6/7, and 8, and overestimate at facilities 5, 12, 18, and 20). Reported emissions almost equally underestimate (facility 4) or overestimate (facility 5) emissions even if there is agreement with measurements on other survey days. Keep in mind that the operator-reported annualized emissions account for both normal and other operations throughout the year. Consequently, the robustness of a top-down vs. bottom-up comparison of an individual facility increases with sample size such

that the measured normal and other operating states are representative of the annual weighted average operating states.

Note that at the five facilities with repeat surveys on different days under normal operations (blue dots at facility IDs 4, 5, 6, 10, and 19 in Figure 5a), the average day-to-day variability in measured emissions for the same facility is 33% (even after accounting for measurement uncertainties). That is, emissions at the same facility vary substantially over time, even on days when the operational status suggests continuous emissions. This implies that intermittency exists beyond the granularity (or

the categories) of the level of reporting above. Nevertheless, as shown in Figure 5b, at the aggregate level of 19 facilities (34 surveys including repeats; number is slightly different from column 5 in Table 1, which separates facilities reported jointly), reported emissions agree with average measurement-based fluxes within 16% irrespective of operating status (Figure 5b right-most column, just outside the $1\sigma$ error). When considering only normal operations, this difference is only 8% (Figure 5b middle column, with $1\sigma$ error). The direct comparison of measurements during normal and other operations at facilities 4, 5, and 6

and 7 indicates that average emissions during other operations are 29% larger than during normal operations for these facilities, although this difference is largely driven by one outlier in facility 4. It is noteworthy that the majority of the randomly timed surveys (10 out of 16 surveys) at facilities 4, 5, and 6 and 7 occurred during other operations. Considering all 20 surveyed facilities, 15 out of the randomly timed 34 surveys were under other operations. As such, an annual extrapolation of only the measurements under normal operations would substantially underestimate annual emissions given the frequent occurrence of

other operations. While accounting for operational status will be key for prioritizing emission mitigation solutions, our results suggest that ensuring a sufficiently large and representative sample size is key for an unbiased estimate of total emissions at the facility-level (repeat surveys) or the regional-level (multi-facility surveys) irrespective of operational status. While the cost of the surveys and the monetary and environmental benefits play a role in designing routine surveys, frequent surveys could ensure the most robust validation.




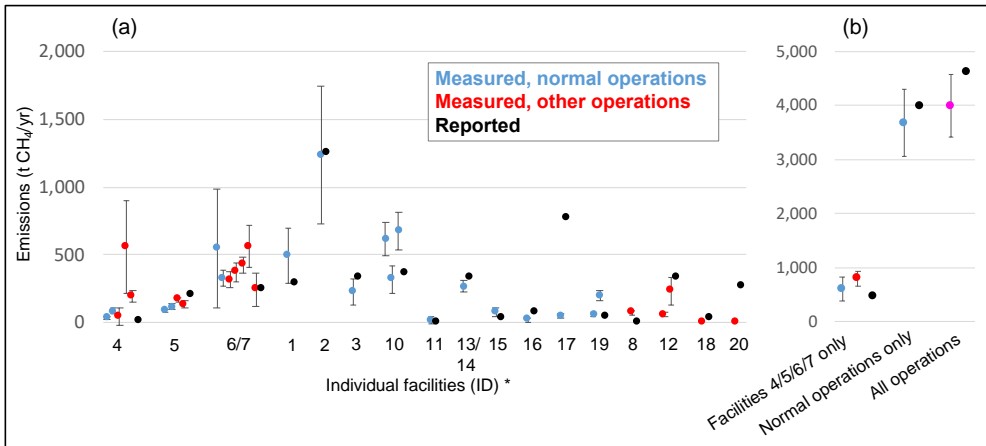

Figure 5. Facility-level $CH_4$ emissions (multiple data points per facility represent repeat surveys on different days). Panel (a): Measured (red and blue) and facility-reported (black) $CH_4$ emission estimates by facility ID number. Panel (b): Total emissions (based on average facility emissions for repeated surveys). The first column includes only facilities 4, 5, 6, and 7 (representing "other" operations). The second column represents all other facilities (representing "normal" operations). The third column represents all facilities collectively (magenta data point). Error bars represent 1σ uncertainties. * Facilities 6/7 and 13/14 were measured separately (and fluxes were added), but operator reported a combined estimate. Facility 9 is not included here because it was reported inactive during 2019 and measured $CH_4$ emissions were negligible (see Table 1).

We further calculated $CH_4$ loss rates, i.e., the measured, annualized $CH_4$ emissions as a fraction of the marketed $CH_4$ over the same period. This was conducted at the field-level (for which gas production data was available; Norwegian Petroleum Directorate, 2021), which includes between one and six individual facilities depending on the field. NOGA (2018) guidelines were used to approximate gas composition to convert total gas production (Norwegian Petroleum Directorate, 2021) to $CH_4$ production. Measured $CH_4$ loss rates range between 0.003% and 1.3%, thus spanning three orders of magnitude. This wide range in loss rates is largely driven by the equally wide range in gas production across the 10 fields, spanning four orders of magnitude. While there is no apparent correlation between absolute emission rates and leak rates, all four fields with loss rates >0.1% each produce <0.15 billion $Sm^3$, and all six fields with loss rates <0.1% each produce >0.5 billion $Sm^3$. Thus, the very small loss rates <0.1% are largely explained by the large denominator (gas production volume). The gas production-weighted average loss rate for the 10 measured fields is 0.012%, but this value should not be considered representative of Norwegian offshore production, and it is very likely a conservative estimate for the full population of Norwegian sites. This is because the 10 measured fields in this study account for 48% of Norwegian gas production, but for only 12% of the total number of producing fields (Norwegian Petroleum Directorate, 2021). In other words, measured fields in this study are strongly biased towards high gas production fields, which in turn explains the relatively small weighted average loss rate.

### 4.4 Outlook

In summary, these results act as a comparison of top-down measurement-based emission quantification, bottom-up facility-specific calculations (similar to the IPCC Tier 3 approaches; IPCC 2006, 2019), and bottom-up IPCC Tier 1 calculations using generic emission factors (used in Scarpelli et al., 2020). As outlined in Sect. 2.5.2., Tier 3 approaches are more rigorous and detailed, applying facility-level emission and activity data to calculate emissions. Results in this study show that there is better agreement between measured data and facility-level reported emissions than more generalized spatially downscaled inventory estimates, as expected. This result emphasizes the importance of facility-level emissions reporting in order to compile accurate national greenhouse gas inventories. This study exclusively considers offshore O&G facilities, adding to the findings from





previous work which found that spatially downscaled inventories may be significantly underestimating $CH_4$ emissions (Gorchov Negron et al., 2020). However, other studies have also observed discrepancies in inventory estimates for onshore facilities (Zavala-Araiza et al., 2021), thus highlighting that Tier 1 inventories can be subject to very high inaccuracy across the O&G sector as a whole.

In this context, it is important that the availability of Tier 3 reported data is increased and more routinely required by regulators and policymakers, and that such data is used to more meaningfully inform overall IPCC emissions scenarios, which may currently contain large underestimates for the offshore O&G sector where only spatially downscaled estimates are available. This represents both a global and field-specific challenge, as individual basins typically comprise multiple operators with potentially different performance standards and reporting frameworks. There is an urgent need for consistent,

internationally agreed standards of best practice, if reported fluxes are to be of value in accurately understanding global emissions from the O&G sector.

Additional measurements are needed to further test and validate global emission inventories. However, collecting such data is labour-intensive and, thus, expensive when using manned aircraft. Slow-moving, lightweight airborne measurement platforms, such as the Mooney aircraft are well-suited to this application, as they allow for much more focussed sampling,

with the ability to densely sample in close proximity to individual O&G facilities. However, future improvements and advances in satellite remote sensing could provide routine datasets to assess facility-level and area-emissions reporting, providing greater spatial and temporal coverage. However, flux measurements in the offshore environment via satellite remote sensing are challenging due to the use of less frequent glint mode observations (for passive near-infrared sensors). Other survey platforms, such as unmanned aerial vehicles (UAVs) also offer potential for $CH_4$ flux quantification from numerous sources

(e.g. Nathan et al., 2015; Yang et al., 2018; Allen et al., 2019; Shah et al., 2020; Shaw et al., 2021). For an interesting overview of $CH_4$ detection technologies for offshore environments, see Carbon Limits (2020). Frequent surveys could lead to measurement-based inventories, similar to that compiled by Gorchov Negron et al. (2020), as efforts continue to quantify emissions and seek to combat global climate change.

## 5 Conclusions

This study reports $CH_4$ fluxes derived from airborne sampling campaigns on the Norwegian Continental Shelf. We conducted 13 flights using the FAAM and the Scientific Aviation research aircraft in July, August, and September 2019.

Measured $CH_4$ emissions were found to range from 2.6 to 1200 t year$^{-1}$ (with a mean of 211 t year$^{-1}$ across all facilities). Mean measured fluxes (as an aggregate of the 21 facilities studied) were 16% lower than equivalent operator-reported data but agreed within 1σ uncertainty. Operator-reported emissions data contains an increased level of granularity concerning operational

emissions and sources, better representing the reported facilities, relative to IPCC Tier 1 data used in the global inventory, making it more closely analogous to IPCC Tier 3 methods. Measured $CH_4$ emission loss rates (as a percentage of $CH_4$ production) ranged from 0.003% to 1.3% across fields, with the wide range largely driven by field-level production volumes, with high-producing fields displaying proportionately lower emission rates. The aggregated comparison with facility-level reported data suggests that measurements and reported data agree within uncertainty, given a large enough sample size to aid

statistical representation. With this in mind, we recommend that similar facility-level reporting is adopted more widely by industry and that reported data is used to more accurately compile national emissions inventories of $CH_4$ relevant to IPCC emissions scenarios. This reporting approach is consistent with the voluntary commitment required for membership in the Oil and Gas Methane Partnership 2.0.

We also compared aircraft-derived fluxes with facility fluxes extracted from a global gridded fossil fuel $CH_4$ emission

inventory compiled, finding that the measured emissions were 42% larger than the inventory for the 21 facilities surveyed (in aggregate). We interpret this large discrepancy not to reflect a systematic error in the operator-reported emissions, which agree with measurements, but rather the representivity of the global inventory due to the methodology used to construct it and the





fact that the inventory was compiled for 2016 (and thus not representative of emissions in 2019). This highlights the need for timely and up-to-date inventories for use in research and policy.

This study also demonstrates the use of airborne sampling to obtain flux snapshots for comparison with inventories and reported data. We found that measurement sampling density, especially in the vertical plane, can dominate sources of uncertainty in aircraft-based flux methods. To reduce uncertainty in flux calculations further using measurement-based approaches, we recommend the use of measurement platforms with a high degree of manoeuvrability.

*Code and data availability.* Data from the MOYA FAAM aircraft campaign is available from the Centre for Environmental Data Analysis (CEDA) archive (https://www.ceda.ac.uk), at https://catalogue.ceda.ac.uk/uuid/dd2b03d085c5494a8cbfc6b4b99ca702. Please note that access to CEDA datasets and resources may require a free CEDA login account. Data from the Scientific Aviation aircraft campaign will also be archived on CEDA. Data can also be requested from the corresponding author.


*Author contributions* AmF curated the data and led the data analysis and manuscript preparation (with contributions from GA, JTS, PaB, JL, JP, AlF, IP and SS). GA was a principal investigator (PI) and was responsible for acquiring funding, project administration, conceptualisation, methodology, supervision, making measurements onboard the FAAM aircraft and manuscript preparation. JTS assisted with data analysis. PrB and PaB made measurements onboard the FAAM aircraft. LH

assisted with data analysis. JRP provided advice and input to the flux analysis. JDL was a principal investigator (PI) and was responsible for acquiring funding, project administration, conceptualisation, methodology, making measurements onboard the FAAM aircraft and writing. SEW provided assistance with data visualisation. PD made measurements onboard the FAAM aircraft. RMP was responsible for project administration. DL was responsible for project administration and conceptualisation. JLF, REF, AlF and MP provided advice and input to the analysis. SJ-BB made measurements onboard the FAAM aircraft.

SAC and MLS made measurements onboard, and analysed the data from, the Scientific Aviation aircraft. TL-C was responsible for project administration. IP configured and ran simulations on the FLEXPART dispersion model. SS was responsible for project coordination, obtaining and analysing emission data from facility operators, data analysis, methodology, conceptualisation, and manuscript preparation.

*Competing interests.* The authors declare no competing financial interest.

*Acknowledgements.* This work was supported by the Climate and Clean Air Coalition (CCAC) Oil and Gas Methane Science Studies (MSS) hosted by the United Nations Environment Programme. Funding was provided by the Environmental Defense Fund, the Oil and Gas Climate Initiative, the European Commission, and CCAC (reference: DTIE19-020). The data used in this publication have been collected as part of the Methane Observations and Yearly Assessments (MOYA) project funded by the Natural Environment Research Council (NERC) (The Global Methane Budget, University of Manchester reference:

NE/N015835/1 Royal Holloway, University of London reference: NE/N016211/1). Airborne data were obtained using the BAe-146-301 Atmospheric Research Aircraft (ARA) and the Mooney Acclaim aircraft. The former was flown by Airtask Ltd and managed by FAAM Airborne Laboratory, jointly operated by UK Research and Innovation (UKRI) and the University of Leeds. The latter was flown and managed by Scientific Aviation. We would like to give special thanks to the Airtask pilots and engineers and all staff at FAAM Airborne Laboratory, as well as the pilots from Scientific Aviation for their hard work in

helping plan and execute successful MOYA project flights. We would also like to thank staff at Kiruna airport, Norway, for hosting the FAAM aircraft during the campaign.





## Appendix A: FLEXPART Dispersion Model: Example Forward and Backward Simulations

Figure A1 shows an example curtain plot for flight C193. Such plots were constructed from the forward simulations of the FLEXPART model for FAAM flights C191, C193 and C197, in order to estimate the modelled plume height. The release of a unit mass from selected rig locations yields 4D FLEXPART output (in e.g. ppb) and provides the basis for interpolation along and below/above the flight track. The derived PBL height is generally consistent with flight data. The forward FLEXPART simulations were based on regional ECMWF winds, which were retrieved specifically for this application. Their domain is [0 E 12E 57N 68N], with 137 hybrid levels. The winds were natively interpolated at 0.1 degrees horizontally. The runs were performed with high temporal resolution, with a synchronization time (internal FLEXPART time step) of 50 seconds. The turbulence in the Planetary Boundary Layer was parametrized with refined horizontal and vertical Lagrangian time scales, represented by the FLEXPART parameters CTL = 40 and IFINE =10 (Pisso et al. 2019). The gridded output resolution is 0.01 degrees with domains containing the flight track and the targeted rigs. The time step of the gridded output for the plumes is 50 seconds (FLEXPART parameter LOUTSAMPLE) averaged over over 1 hour (FLEXPART parameters LOUTSTEP and LOUTAVER set to 3600).

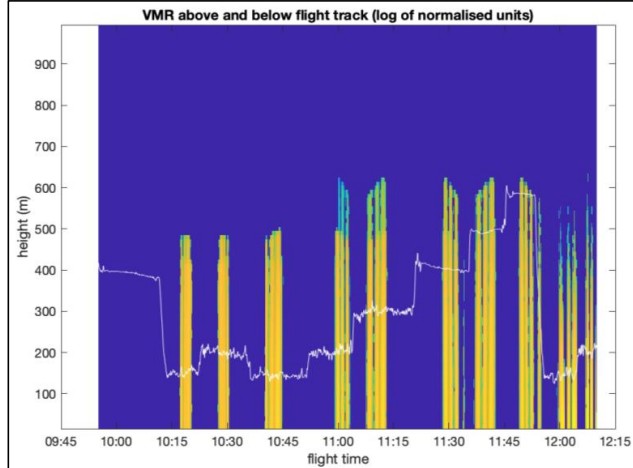

Figure A1. Example curtain plot for the forward FLEXPART simulation for FAAM flight C193, used to estimate modelled plume height. The white line denotes the flight altitude and the shaded area denotes the logarithm of the normalised volume mixing ratio of $CH_4$ (column containing each measurement).





Figure A2 shows an example modelled footprint for flight C193, based on backward trajectories simulated by the FLEXPART model. These simulations were conducted for FAAM flights C191, C193 and C197. The calculation of retroplumes (or "footprints") for each measurement point along a flight track allows the identification of oil and gas platforms linked to individual peaks detected in the time series of measured $CH_4$. The magnitude of the retroplume is proportional to the time averaged spent by trajectories in the corresponding grid cell.

745

Figure A2. Example snapshot of the calculated FLEXPART footprint for FAAM flight C193, used to aid source identification for measured $CH_4$ enhancements. The flight track is coloured by measured $CH_4$ mixing ratio (right colour bar; ppbv). The shaded area denotes the vertically integrated retroplume (left colour bar; s m$^2$ kg$^{-1}$). The red triangles represent the locations of the nearby offshore O&G facilities.





**Appendix B: Aircraft Observational Data**

Figure B1 shows the flight track of FAAM flight C193, which took place on 30th July 2019, along with nearby offshore O&G facilities. Figure B1a shows the measured wind speed and direction (shown as arrows) and Figure B1b shows the measured

780    $CH_4$ mole fraction. The FAAM data showed $CH_4$ enhancements above background which typically lied between approximately 2 and 13 ppb. However, much larger enhancements were seen in region 2 overall, with a maximum of 99.3 ppb above background. A maximum of 8.9 ppb was observed in region 1. This was as expected as the facilities in region 2 were known to produce substantially more oil and gas compared to region 1, as seen in Figure 1 in Sect. 2.3 of the main paper.

785

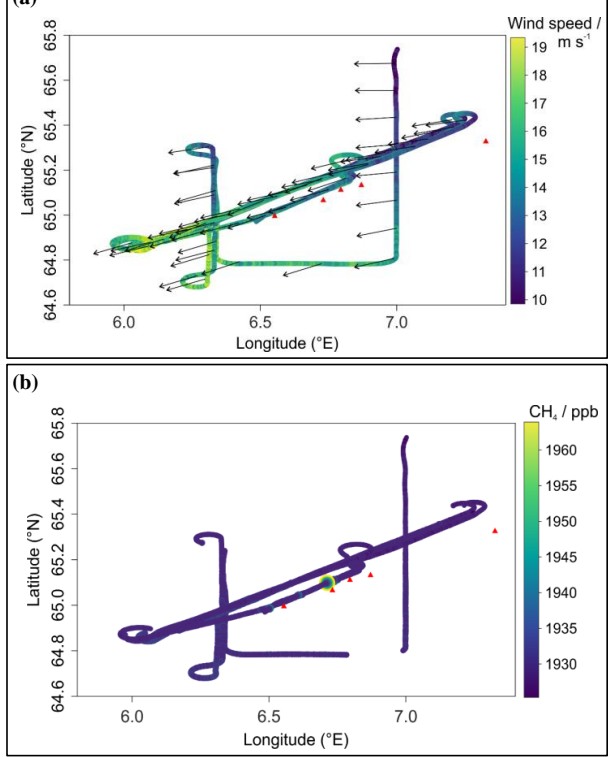

Figure B1. Flight track for flight C193, (a) colour-coded by the wind speed, with arrows denoting the wind direction over the course of the flight, and (b) colour-coded by the $CH_4$ mixing ratios. The red triangles represent the locations of nearby offshore O&G facilities.







Figure B2 shows the altitude-longitude projection of the vertically stacked transects from flight C193, as an example. During

the flight, seven transects were flown downwind of the offshore facilities, with spacing of between 50 and 100 m between

each transect. Flights C191 and C197 comprised seven and fourteen vertically stacked legs, respectively, with spacing of

between 50 and 100 m between each transect.



Figure B2. Altitude-longitude projection of the vertically stacked
downwind transects conducted in flight C193, coloured by the $CH_4$
mixing ratios.

Figure B3 shows an example of mapped $CH_4$ mixing ratios for a Scientific Aviation flight survey which took place on 21st

August 2019. The $CH_4$ enhancements above background were generally higher than those observed in the FAAM flights,

typically lying between 10 ppb and 50 ppb, due to the closer proximity of measurement to the facility sources.


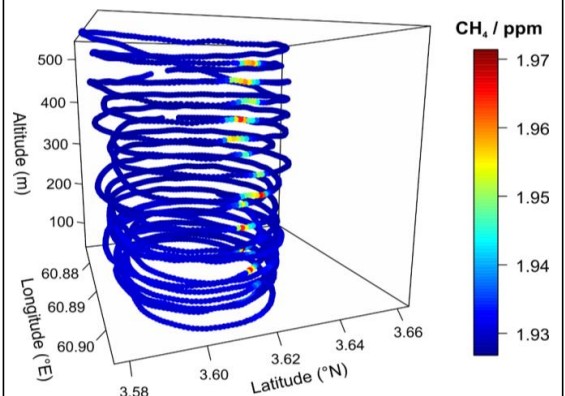



Figure B3. 3-dimensional map of the flight pattern of the Scientific
Aviation aircraft sampling a $CH_4$ plume from an offshore O&G facility.

During all flight surveys, background concentrations were consistently invariable relative to observed downwind

enhancements (see Figures B1 and B3) by virtue of the remote maritime sampling environment and absence of significant

nearby pollution sources This aided detection of any $CH_4$ plumes downwind of facilities. Overall, wind fields were stable over

the course of the flight surveys, facilitating the mass balance methodology described in Section 3.1 of the main paper. Across

all FAAM and Scientific Aviation flights, wind speeds varied between 1 and 19 m s$^{-1}$. Observed wind directions were also

consistent during the flights, with FAAM flights C191 and C197 experiencing southerly winds, and flight C193 experiencing

north-easterlies (as shown in Figure S3).



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
