# Peer review of "Quantification and assessment of methane emissions from offshore oil and gas facilities on the Norwegian Continental Shelf"

_Atmospheric Chemistry and Physics, 2021_

## Author Comment (AC1)

**Authors' response to reviewers for ACP 2021-871**

The authors would like to thank the reviewers for their thorough and constructive reviews, and their insightful comments and suggestions. Below is a breakdown of all reviewer comments (*black italics*) with corresponding author responses (red text).

**Reviewer 1**

Foulds et al. measured methane (CH4) from aircraft and determined emissions from 21 oil and gas facilities in the North Sea oil region of Norway. They found emissions agreed with self-reported emissions from oil and gas companies, but were higher than a downscaled CH4 global inventory from three years prior to the study.

Overall, I thought this was a good study worthy of publication in this journal. The study is scientifically significant, especially when considering the greenhouse gas footprint of fuel choices. The scientific methods are valid and have been used frequently in the past. I only have a few comments I'd like to see addressed before I think this paper is ready for publication.

The authors would like to thank the reviewer for their positive comments concerning the scientific significance of the study, and for their useful and informative review.

**1) Specific comments**

Introduction: by the end of the manuscript, the authors spent a lot of time referring to Tier 1 and Tier 3 inventories. I think this warrants a paragraph in the introduction defining and discussing the different tiers. How are they constructed? When are they used?

We first introduce Tier 1, Tier 2 and Tier 3 methods in section 2.5.2 of the manuscript (entitled "Global inventory of CH4 emissions from oil and gas exploitation"). Here, we include a discussion of what the different approaches mean. As this is the first place we introduce these tiers, we feel that this is the most suitable place to introduce and describe them. This then means that the reader is able to understand what is meant by "Tier 1" and "Tier 3" approaches later in the text. We feel that, to introduce them any earlier in the manuscript would not be appropriate, as it would not make sense within the introduction.

Section 4.1: Based on Figure B1a, it seems like the wind speed perpendicular to the flight track would be highly uncertain, since several transects were nearly parallel with the wind. Please explain how the uncertainties for C193 do not dominate the uncertainty calculation. Also, what is the estimated uncertainty for the wind direction measurement?

The reviewer is correct - the wind direction is just a few degrees perpendicular to the flight track in this example. In our experience, this might be expected to demonstrate quite large flux uncertainty. However, it is not the wind direction that matters for flux uncertainty, rather the variability in the wind vector component that is perpendicular to the sampling frame. Any variability would be magnified here, due to the small angle between the flight track and the wind. However, the wind vector (speed and direction) is quite consistent in this case, as shown in the figure. The wind-specific flux uncertainty component for flight C193 was calculated to be between 5% and 7% across the three plumes identified and shown in the figure. Lines 383-384 of the manuscript discuss that uncertainty associated with wind variability is typically <10%, which remains the case for flight C193. As an aside, in our field experience, it is unusual to see such consistent wind speed and direction. In many other case studies we've worked on over several years, such a small angle perpendicular to the sampling frame would lead to much higher uncertainties with even a small variability in wind speed. However, in this case, environmental conditions were quite static during sampling.

line 387: this seems like the lower plume extent uncertainty would be greater for the Scientific Aviation flights than for the FAAM flights. If you already assume the plume is well mixed at the top of the boundary layer for the FAAM flights, it must also be well mixed below the lowest altitude of the aircraft.

The reviewer is correct. We do assume that the air is well-mixed down to the surface. We include the layer below the aircraft (where we do not have sampling to confirm the presence of a plume) in our flux uncertainty budget. The FAAM aircraft typically had a higher minimum altitude than the Scientific Aviation aircraft (124 and 20 meters above sea level for the FAAM and SA aircraft, respectively). However, most of the plume height uncertainty is associated with the altitudes above the aircraft (with Scientific Aviation flights having lower uncertainties, due to the ability of the smaller aircraft to get above the plume). This therefore means that the uncertainties in the plume mixing remain higher for the FAAM flights than for the Scientific Aviation flights, as discussed in the text.

line 443: I disagree with the authors conclusions here. Rather than claiming it is not of value to compare measurements to a spatially downscaled inventory, I think you can use measurements to instead question the value of the inventory itself. What the authors have found may in fact be evidence that a national inventory should not be spatially downscaled in this way, but that is different than saying a comparison to measurements have limited value.

We understand the reviewer's concern, and to claim a limited value is a vague conclusion to make. We have therefore rephrased the text in the manuscript to the following: "We include the result here to make a value point, which is to say that downscaling of inventories can lead to significant discrepancies at the scale of oil and gas facilities, such as those studied here.".

Figure 4: since the authors are simply fitting y vs. x, there is no reason the data plotted on the y-axis need to be upscaled to t/yr. Why not just keep them as kg/s or whatever the units of the flux measurement are?

The inventory and emissions data reported by the platform operators are in units of t/year, as shown in Table 1. We have used units of t/year for consistency, a) throughout the paper, and b) with the way fluxes are reported in the inventory and by the operators. We would like to retain consistency in this case. We have added a sentence in the manuscript to account for this on **line 434**, which reads: "Measured emissions fluxes are reported in units of t year-1, in order to ensure consistency with the units used in the emission inventory and facility-level reported data.".

**Figure A1: it seems the color legend is missing from this graph**

A colour scale has been added to Figure A1, and the revised figure has now been added to the manuscript. A log scale is used in Figure A1, simply for better contrast in the border of the plume. The absolute values are arbitrary, since the source emission rate that would be needed for a realistic concentration simulation is the unknown, but this is irrelevant for the visual inspection of the plume extent.

Figure A2: Is the FLEXPART footprint taken from a spot on the flight track? Because it looks like it is offset further downwind from the flight track.

We thank the reviewer for noticing this. This was found to be a graphical error, whereby the axes of the footprint were shifted up during the rendering of the plot. This has been rectified, and a revised version of Figure A2 has now been added to the manuscript.

line 393: the phrase "clearly not a robust comparison" seems subjective unless the authors show day-to-day variability in emission, which at this point in the paper, they haven't. I suggest "likely is not" or something like that.

We agree that this is subjective. The text has been amended to the following (**line 398**): "Scaling in this way is likely not to be a robust comparison as it cannot account for any variability in day-to-day facility operations throughout the year.:

line 647: I prefer the sentence in the abstract that mentions "all 21 facilities". I would add that to the conclusions section as well.

We agree. This would provide consistency within the manuscript. The text has therefore been amended in the conclusions section. It now reads "Measured  $CH_4$  emissions were found to range from 2.6 to 1200 t year-1 (with a mean of 211 t year-1 across all 21 facilities).

**line 652: is this across "fields" or across all facilities?**

This is across all facilities (with fields being implicit). We have changed the text in the manuscript to make this clear to the reader. It now reads as follows: "Measured  $CH_4$  emission loss rates (as a percentage of  $CH_4$  production) ranged from 0.003% to 1.3% across all facilities, with the wide range largely driven by field-level production volumes, with high-producing fields displaying proportionately lower emission rates."

2) Technical comments

line 69: add "the" before "UK"

This has been added to the text.

line 71: remove comma after "et al."

This has been removed from the text.

line 81: add period after "et al"

This has been added to the text.

line 114: don't hyphenate "in-flight"

The text has been changed. The hyphen has been removed and it now reads "in flight".

line 267–268: annual is mentioned twice in this sentence

The sentence has been amended. It now reads "Operators on the Norwegian Continental Shelf are required to submit annual CH4 emissions data to the Norwegian Environment Agency every year."

line 300 and elsewhere: suggest "data ... were"

This has been amended at this point, and throughout the text.

line 386: suggest starting sentence with "However, there is also some additional uncertainty..."

This has been amended in the text. The sentence now reads "However, there is also some additional uncertainty if the bottom of the plume cannot be sampled", which we hope reads better than the original.

line 660: is "compiled" a typo? It doesn't make sense to me.

Yes, this is a typo. The word "compiled" has been removed from the text.

**Reviewer 2**

The authors provide measurements of methane emissions from 21 offshore facilities using two different aircraft-based platforms. The findings basically show the utility of such measurements to validate national emission inventories and improve emission estimation practices, including national reporting guidelines. The main implication that the authors communicate is the need for more measurements; however, the results can also be used to help understand the root cause of the emissions and to help improve ways in which the industry reports their emissions and operations. Overall, the paper is well-written and only needs minor modifications.

Although the paper does a very nice job of providing information that can validate inventories, I think the authors can provide more information on characteristics of the offshore facilities that can be used to identify the root cause of the emissions. For example, even a simple differentiation of facilities by the type of hydrocarbon being produced could be a helpful start. Moreover, I think the paper points to some opportunities in which emissions can be reported better by industry and if so, these points should be made more explicit. Below are some detailed suggestions.

*Line 30: It would be good to add the word "only" in front of "16%" to emphasize that 16% is small.*

We agree that this would add emphasis to the small difference between the aircraft-measured fluxes and the operatorreported data. The word "only" has been added, and the text now reads: "...with mean aircraft-measured fluxes only 16% lower than those reported by operators."

*Line 38: What is meant by "this"? Is "this" "measurements of temporal variations" or "knowledge of facility operational status over time"?*

We agree this was unclear. We also think that the definitive way we phrased this sentence was also a little ambiguous. So we have changed the sentence to: "Future surveys of individual facilities would benefit from knowledge of facility operational status over time."

*Line 39: How big of a sample is sufficiently large? Knowing how many times a facility should be measured would help regulators. How do we know we have a representative sample?*

A fully representative sample must capture the distribution of methane emissions across the entire population of offshore platforms, which also must represent different facility types (facility ages and variable facility designs, as well as production volume). Clearly, the more facilities sampled, and the more often, the better, but there are significant constraints in both expense and time that are challenging to mitigate. Given this, the quite vague reference to the need for a "sufficiently large sample size" in the original manuscript clearly does not give the reader enough information to make an informed judgment to guide future surveys that may serve emissions science following our work. We have now contextualised this statement to capture the need for factors above and commented further on what our study does and does not do within the logistical constraints of resource-intensive aircraft fieldwork.

Moreover, achieving truly randomised sampling is problematic as it is more efficient to sample co-located facilities when using aircraft i.e. to sample multiple platforms in the same local field, than it is to sample random facilities distributed across a whole region. For our study, this means that we have an intensive survey of a local field, but that our results are specific to the Norwegian Sea fields and cannot be extrapolated to other production fields.

We would recommend conducting annual surveys by field, as emissions are reported annually. We have now added a discussion about this in the abstract of the revised manuscript.

Line 40: Specify "aircraft" in front of "measurement approaches", unless the point here is that operators use any measurement approach.

We have amended the text to make the sentence less ambiguous. It now reads: "In summary, this study demonstrates the importance and accuracy of detailed, facility-level emission accounting reporting by operators and the use of airborne measurements approaches to validate bottom-up accounting."

*Lines 56-57 and rest of paragraph: How much of the methane emissions from the O&G sector does offshore O&G production represent? This type of information would be valuable somewhere in this paragraph.*

We added the following to contextualize the study despite the little data available so far: "There have been limited numbers of studies focussed on emissions from offshore O&G production, relative to onshore facilities (EIA, 2016a). The current quantification of emissions from offshore facilities therefore often relies on bottom-up approaches that use activity data and emission factors to derive emissions from a sub-set of sources, and extrapolation to estimate a total emission. However, emission factor calculations rely on representative knowledge of all emission sources, with the potential for systematic error. The International Energy Agency (IEA) Methane Tracker bottom-up estimate of the offshore share of global O&G related methane emissions is 20% (IEA, 2021)."

Note that the IEA reference added to the text is the online IEA Methane Tracker database: https://www.iea.org/articles/methane-tracker-database

*Lines 63-65: Are these studies offshore or onshore studies? How do the authors expect the offshore emissions to be different from onshore emissions?*

These studies are collectively offshore and onshore studies. We have amended the text now make this clear. It now reads: "The studies that have taken place (for both onshore and offshore facilities) have consistently reported inventory underestimates of CH4 and non-methane volatile organic compounds (NMVOCs) from O&G extraction..."

*Line 73: What is the ACCESS campaign? Spell out ACCESS.*

We agree that spelling this out would provide useful information to readers. The text has been amended to include what the "ACCESS" acronym stands for. The text now reads: "As part of the ACCESS (Arctic Climate Change, Economy and Society) campaign..."

*Line 103-104: A short sentence here on how the FAAM platform is much larger than the Scientific Aviation platform would be helpful here. Also, add "two" in front of "aircraft platforms".*

The text has been modified to include a sentence about the sizes of the two aircraft. We have added the following text: "In Sect. 2.1, we describe a larger Bae-146 aircraft, which is a 4-engine passenger jet, modified as a flying laboratory. In Sect. 2.2, we describe the smaller, single engine Scientific Aviation Mooney aircraft.". In response to the second part of the comment, the text has also been amended to "we describe the flight surveys, the two aircraft platforms and instrumentation used...".

Figure 1: It would be helpful if the oil and gas production was presented in such a way that we can distinguish between oil and gas (and even condensate or mixed), in addition to production amounts.

Figure 1 has been amended in the revised submission. The field type (oil, gas, condensate, or mixed) is now illustrated by the colour of data points, whilst oil and gas production volume is indicated by the size of the data points.

Line 188: The production numbers in Figure 1 are 3-4 orders of magnitude lower compared to what's in Figure 2. Is this a typo? And importantly, are these production amounts really large? What are the amounts of oil and gas produced in other regions (e.g., Gulf of Mexico)?

We thank the reviewer for noticing this. The reviewer is correct; there is an error in the production numbers shown on the colour bar in Figure 1. This has been corrected, and the amended version of Figure 1 is now included in the manuscript. We have called the production numbers large because they are large for the region that we were studying. We are looking at individual facilities here, so we can't compare at field-level with the Gulf of Mexico.

*Line 198-199: Were the operators broadly aware of the measurement study, even though they did not know when the measurements were happening?*

Yes, operators were aware of the measurement study. We have modified the text to account for this, adding the following sentence: "However, operators were aware of our study, but not the time or the sampling pattern of the flights.".

*Figure 2: These oil & gas production numbers are 3-4 orders of magnitude larger than those in Figure 1.*

See response to the comment about line 188 above.

Figure 2: Could the facility numbers be shown here as well? Also, what is the difference between the cluster of red facilities and the rest, which are mainly blue? It seems that there is clearly a spatial pattern, possibly governed by geology and the produced hydrocaron (i.e., gas vs. oil).

We thank the reviewer for this comment. However, there is an NDA in place with operators who shared non-public, temporally specific activity data to improve insights for this study. Putting facility numbers on the figure would link them (and our flux results) to the geographical location of the facilities. Showing facility locations will violate the NDA, meaning that we are unable make this addition to the figure.

*Line 261-262: How do the Norwegian guidelines compare with the IPCC guidelines and guidelines from other countries (e.g., U.S.)?*

The Norwegian guidelines are similar to Tier 3 reporting in the IPCC guidelines and we have added the following sentence to the manuscript to account for this: "This level of reporting is similar to Tier 3 IPCC guidelines (see Sect. 2.5.2.".

We have described the Norwegian approach in detail because that is the focus of this study. However, we have added a sentence which accounts for the fact that different countries and operators within basins may use different reporting procedures: "It should therefore also be noted that different countries and operators are likely to use different reporting procedures."

*Line 268-269: Sentence is not complete.*

This has been corrected in the text. The sentence is now complete and reads as follows: "The CH4 emissions are reported for individual sources and sub-sources (e.g. primary vent seals for centrifugal compressors and incomplete combustion in flares)."

Line 272: A list of all potential sources would be helpful here, especially for mitigation

The top-down method employed in this paper does not distinguish among these sources. For further detail, please refer to the referenced NEA and Usdal studies.

Equation 1: Define A and B and zmax.

We have amended the text to include the definitions of A, B and  $z_{max}$ . The text now reads: "where, F (g s-1) is the flux for the emission source, A and B are the horizontal boundaries of the plume,  $z_{max}$  is the maximum plume height, Cij is the dry mole fraction of CH4 at each point in the plume, C0 is the representative background dry mole fraction of CH4, nair is the molar air density and ULij is the wind speed perpendicular to the reference measurement sampling plane."

*Line 327-328: How many vertical layers are considered at the sites? I recognize that this will be a range.*

Yes, the reviewer is correct in that there is a range of vertical layers. Across all three flights, the number of layers varied between 2 and 6 layers, based on the number of runs and the spacing between them.

Line 333-335: How different are the mixing latitudes obtained using these different approaches?

We assume that the reviewer means mixing altitudes in this comment. In this respect, the mixing altitudes ranged between approximately 360 and 700 metres above sea level. This range was due to the fact that the measured boundary layer meteorology varied from day to day. Also, in some cases, we were able to get a much better constraint on the mixing height, as we sampled at altitudes where no plume was detected. In other flights, there were altitudes where we didn't have sampling, thus meaning a poorer constraint on the plume mixing height. However, we include any areas where we didn't sample in the uncertainty budget, in order to account for this. This is discussed in Sect. 4.1 of the paper.

Line 397: Do the Norwegian guidelines require reporting of emissions at the annual level? In such annual estimates, there would have been estimates of the number of emission events per year.

With regards to the first part of the comment, as discussed in Sect. 2.5.1, Norwegian guidelines do require reporting of emissions on an annual basis. (see sentence: "Operators of facilities on the Norwegian Continental Shelf are required to submit annual CH4 emissions data to the Norwegian Environment Agency every year."). In Norway, some intermittent emission events, such as from oil offloading to tankers, are tracked and reported. Another example is methane emissions from flaring, which are partially measured via the flare volume by means of flare meters. In this study, consistent activity data at the required temporal resolution (hourly or more granular) was only available qualitatively (as described in section 4.3).

Line 438: I'm not sure if it makes sense to say these measurements are "outliers". They are still a part of operations (i.e., turbine maintenance). Instead, I think it points to how the inventory guidelines could be improved to be more consistent with operations.

We agree that these measurements should not be referred to as "outliers", as this implies that they are not valid results. In order to address this, we have amended the text to the following: "Figure 4b shows two facilities that do not fit the pattern (within uncertainty), with reported fluxes of 270 and 780 t year-1."

We also agree with the reviewer, in that these two facilities demonstrate the need for improving inventory guidelines and have added the following text to the manuscript: "These results from these two facilities demonstrate how inventory guidelines need to be improved to ensure more consistency with operations.".

Figure 4: It would be good if these plots could indicate the operation or activity that the emissions are coming from. Above, for the outliers, cold-venting and turbine maintenance are mentioned. But what is happening at the other measured facilities?

This is discussed in Sect. 4.3 (and is illustrated in Figure 5), where we link our reported fluxes to operations at each facility. The purpose of figure 4 is just to show potential correlation between measured and reported fluxes. The discussion of activities, etc. is given later in the paper.

Figure 4: Could we get more information on the facilities? For example, are they mainly producing oil or gas?

**Please see the response to comment above.**

Line 470: How do you know the data points in the blue boxes are outliers? How are "outliers" defined? Visually, the left data point (around 300 t/year) doesn't really look an outlier to me.

As discussed in an earlier comment, we agree that it was improper to call these two data points outliers. There are of course, valid results, and not anomalies. We have modified the text (see earlier comment) and the caption of Figure 4 (now reading: "The data points in the blue boxes in Figure 4b) represent two facilities which do not fit with the general pattern, with near-zero measured fluxes"), to account for this.

*Line 483: What is the difference between Facilities 6 and 7? Could this large difference have been predicted?*

Unfortunately we do not have information at the level of detail needed to answer this question for these specific facilities.

**Line 494: Would the facilities measured here be considered deep or shallow water facilities?**

The water depth for the surveyed offshore production fields ranged between 100 and 370 metres [internal communication]. However, there is little reason to expect water depth to have any systematic correlation with the magnitude of CH4 emissions, at least over the range of depths surveyed here. The more important factors driving methane emissions are likely facility design, facility construction, facility age, maintenance procedures, production scale/scope, and the presence of any methane mitigation technologies. We refer the reviewer to the report below, description of CH4 emissions related to which provides а gas production in Norway: https://www.equinor.com/content/dam/statoil/documents/sustainability-reports/greenhouse-gas-and-methaneintensities-along-equinors-norwegian-gas-value-chain-2021.pdf

*Line 504: Which inventory for 2019? The scaled Scarpelli et al inventory? If so, please specify.*

The reviewer is correct, this is the scaled Scarpelli inventory. The text has been amended to specify this. It now reads: "Repeating the analysis above, using the scaled Scarpelli inventory, we find that total measured emissions were 52% higher than the inventory for 2019."

Line 557: Is it reasonable to assume that the errors are normally distributed? Also, are the sample really independent? There are likely to some similarities among facilities with the same operator and in the same basin. And there are likely other relevant factors such as geology, production, and technology.

The errors around the means of the facility-level emission estimates are the result of error propagation from individual parameters (such as measured wind speed and direction and upwind/background and downwind CH4 concentrations. These errors are largely the result of measured variability, which is normally distributed, hence the assumption of total errors being normally distributed. It is indeed not possible to empirically verify the stated assumption that all samples are independent. However, as described in the methods, the sample locations have been selected to include a spectrum of high and low producing facilities as well as multiple geographies with the Norwegian North Sea. Apart from this, the sampling was random, and repeat samples were taken on separate days. Thus, we avoided any dependence to the best of our abilities. While the stated assumption may be imperfect, there is no data we're aware of that could be used to quantitatively inform how any of the potentially dependent factors affect emissions.

*Line 564: a missing "to" in front of "lead"*

**This has been amended in the text.**

*Line 564: Can we say something about how many times per year a facility should be sampled?*

A year-on-year validation that is analogous to the frequency of regulatory reporting, and on the typical timescale of compiled global inventories, would require that facilities are representatively surveyed on an annual basis to provide a direct comparison between annual reporting requirements and a measurement-based assessment. Clearly, this is not always feasible due to logistics and cost. As discussed in response to an earlier comment, we have now added this guidance in the new abstract of the paper.

*Line 581-582: There may be a need to reconsider the use of the term "normal operations". Many of the "other" operations such as maintenance are a part of the normal activities on a platform and may be better viewed as routine.*

We agree with the reviewer, and have reworded the key sections as described below (including other instances of "normal" used throughout the manuscript), exchanging the term "normal operations" with "primary operations". We believe that this is a better choice than "routine operations" since routine doesn't distinguish between intermittent and quasi-continuous methane sources (the latter term is, however, a cumbersome alternative to "primary").

Section 4.3: "The first cluster (IDs 4-7) contains facilities for which measurements were available for facilities under both normal facilities "primary" operations and "other" operations. Primary operations are defined as operations which are central to the production of hydrocarbons and which emit methane almost continuously in the context of this study. Other operations are define as temporal deviations from the normal primary operations (based on operator reports received upon request post-campaign), which may increase or decrease snapshot emission estimates relative to annualized inventories (see details below)."